# Numerical Study of Nanoparticle Deposition in a Gaseous Microchannel under the Influence of Various Forces

**DOI:** 10.3390/mi12010047

**Published:** 2021-01-01

**Authors:** Fubing Bao, Hanbo Hao, Zhaoqin Yin, Chengxu Tu

**Affiliations:** Institute of Fluid Measurement and Simulation, China Jiliang University, Hangzhou 310018, China; dingobao@cjlu.edu.cn (F.B.); steveyhao@163.com (H.H.); tuchengxu@cjlu.edu.cn (C.T.)

**Keywords:** microchannel flow, rarefied gas, nanoparticles deposition, thermophoresis, Brownian force

## Abstract

Nanoparticle deposition in microchannel devices inducing contaminant clogging is a serious barrier to the application of micro-electro-mechanical systems (MEMS). For micro-scale gas flow fields with a high Knudsen number (*Kn*) in the microchannel, gas rarefaction and velocity slip cannot be ignored. Furthermore, the mechanism of nanoparticle transport and deposition in the microchannel is extremely complex. In this study, the compressible gas model and a second-order slip boundary condition have been applied to the Burnett equations to solve the flow field issue in a microchannel. Drag, Brownian, and thermophoretic forces are concerned in the motion equations of particles. A series of numerical simulations for various particle sizes, flow rates, and temperature gradients have been performed. Some important features such as reasons, efficiencies, and locations of particle deposition have been explored. The results indicate that the particle deposition efficiency varies more or less under the actions of forces such as Brownian force, thermophoretic force, and drag force. Nevertheless, different forces lead to different particle motions and deposition processes. Brownian or thermophoretic force causes particles to move closer to the wall or further away from it. The drag force influence of slip boundary conditions and gas rarefaction changes the particles’ residential time in the channel. In order to find a way to decrease particle deposition on the microchannel surface, the deposition locations of different sizes of particles have been analyzed in detail under the action of thermophoretic force.

## 1. Introduction

As one of the most promising technologies in this field, the microfluidic system has attracted widespread attention in recent years. The transportation, diffusion, and deposition of aerosol particles in microchannels are the common gas–solid two-phase flow problems in MEMS devices, such as microreactors [1], filters [2], and micromixers [3].

The micro-scale gas flow field in a microchannel with a high Knudsen number (*Kn*) begins to deviate from the thermodynamic equilibrium due to insufficient molecular collisions. *Kn* is a dimensionless parameter of the relative sizes of the gas mean free path (*λ*) and the characteristic scale of the microchannel geometry. When 10 ≤ *Kn* ≤ 10, the flow regime is in the transition regime [4]. In this case, the conventional Navier–Stokes(N-S) equations whose constitutive relations are linear in terms of thermodynamic forces are not suitable to be used to solve flow field problems with a high deviation from equilibrium. Sharipov and Seleznevsuthe [5] completed the range of the parameters which influenced rarefied gas flows through a capillary, such as the capillary length, the Knudsen number, the pressure, and temperature drops on the capillary ends. It is an interesting subject to study the rarefied gas slip flow problem in microchannels over recent decades and years [6]. Sharipov studied the rarefied gas flows through microchannels of different forms such as circular capillaries [7], rectangular microchannels [8], and elliptic cross-section pipe, etc. [9]. In order to maintain computational efficiency, Burnett equations which are a set of extended dynamic equations keeping the second-order terms in Chapman–Enskog expansion have been used to solve the flow field issue in the transition regime. There are several variants of Burnett equations proposed in the literature, such as the original Burnett equations, the conventional Burnett equations, the augmented Burnett equations et al. [10,11]. Lots of studies have been derived about the Burnett equations and their validity or instability. Aishwarya et al. [12] solved the conventional Burnett equations to investigate the gaseous flow for a low Mach number in a long microchannel. Agarwal et al. [13] provided numerical solutions to assess the accuracy and applicability of the Burnett equations. The results confirmed the equation is unconditionally stable and has second-order accuracy, which is verified in the compressible pressure-driven Poiseuille flow field. The augmented Burnett equations are always stable in both one- and two-dimensional numerical tests and maintain the accuracy of the conventional Burnett equations in the continuum transitional regime [14]. Xu et al. [15], Bao et al. [16], and Singh et al. [17] applied the augmented Burnett equations to gaseous flow through the Poiseuille and Couette flow fields to obtain the flow and heat transfer characteristics. These results have been proven to match the direct simulation Monte Carlo (DSMC) solution well.

The flow field and forces acting on the particle dispersion and deposition processes in microchannel devices will induce serious problems such as contaminant clogging and abrasion [18,19,20]. Some studies considering slip boundary or rarefied effects of gas-particle flow in microchannels have been performed. Mohajer et al. [21] numerically studied a micro-spherical particle in the slip flow regime and found that a large slip on the wall increases the convection along the surface. Kishore and Ramteke [22] studied the convective heat transfer between spherical particles under slip boundary conditions and showed that increasing the slip parameter would cause the average Nusselt number to decrease. Mu et al. [23] numerically simulated the sedimentation process of particles under the effect of micro-scale rarefied gas, and the research showed that effective viscosity decreased with the growth of the Knudsen number; the particle trajectory was also affected. Unlike conventional particles, micro-forces (such as van der Waals’ force, thermophoretic force, and Brownian force) acting on nanoparticles cannot be ignored. Islam et al. [24] presented the deposition of nanoparticles in lung branches and investigated the deposition positions of particles with different sizes under the action of drag force, Brownian force, and Saffman lift force. Lu et al. [25] proposed that thermophoresis was the main factor of particle deposition in the thermal boundary layer and it had a greater influence on the deposition rate for microparticles. MacGibbon et al. [26] experimentally studied the deposition process of particles in a tungsten low-pressure chemical vapor deposition reactor and found that Brownian motion dominated when particle size was less than 0.01 mm. Thermophoresis dominated while the range of particle sizes varied from 0.1 to 1 mm, and gravity/inertia forces dominated when the particle size was greater than 10 mm.

A suitable solution for gas-nanoparticles in a microchannel should be able to settle the problems of rarefied gas non-equilibrium effect, slip flow effect, and the complex forces acting on the nanoparticles. Despite many studies that have been carried out on particle deposition in channels, there are still some that may be usable in the microscale. However, previous studies have not comprehensively considered micro-nano-scale and rarefied gas effects on particle motion. Therefore, in this paper, the Euler–Lagrangian method has been used to simulate the nanoparticle transport process in a microchannel with *Kn* > 0.1. The rarefaction effect on the gas flow field will be solved by the Burnett equations with slip boundary conditions, and complex forces action on nanoparticles will be analyzed using a kinematic equation adding the Brownian and thermophoresis forces. The main goal of this study is to discover the principle of nanoparticle deposition on the microchannel surface. We will particularly explore the effects of slip boundary conditions, particle sizes, and temperature gradient on particle motion in the transition flow regime.

## 2. Governing Equations

When *Kn* is in the range of 0.1 to 10, the gas flow is in the transition regime. The rarefaction effect becomes a significant influence on the flow field in this flow regime. In this paper, the flow in the microchannel at *Kn* = 0.11 is in the early transition zone. *Kn* of the fluid flow is defined as:(1)Kn=λ/L
where *λ* is the gas mean free path and *L* is the characteristic scale of the microchannel geometry.

The flow field continuum hypothesis is valid but the thermodynamic equilibrium is broken down in this case. As a set of higher-order continuum equations, the Burnett equations with the slip boundary condition can be used to account for the flow field rarefaction effect in the microchannel. The augmented Burnett equations in the transition flow regime can be written as [27]:(2)$$\frac{\partial Q}{\partial t}+\frac{\partial E}{\partial x}+\frac{\partial F}{\partial y}=0,$$
where,
(3)$$Q=\left[\begin{array}{l} \rho \\ \rho u\\ \rho v\\ e_{t} \end{array}\right],E=\left[\begin{array}{l} \rho u\\ \rho u^{2}+p+\sigma_{11}\\ \rho uv+\sigma_{12}\\ \left(e_{t}+p\right)u+\sigma_{11}u+\sigma_{12}v+q_{1} \end{array}\right],F=\left[\begin{array}{l} \rho v\\ \rho uv+\sigma_{21}\\ \rho v^{2}+p+\sigma_{22}\\ \left(e_{t}+p\right)v+\sigma_{21}u+\sigma_{22}v+q_{2} \end{array}\right].$$
among them, *ρ* is the density, *p* is the pressure, *u*, *v* are the velocity components, and *e_t_* is the total energy per unit mass,
(4)$$e_{t}=\rho C_{v}+\frac{1}{2}\rho \left(u^{2}+v^{2}\right)$$
where *C_v_* is the constant volume-specific heat, and *σ_ij_* is the viscous stress tensor and *q_i_* is the heat transfer rate. The expressions are:(5)$$\sigma_{ij}^{\left(2\right)}=\frac{\mu^{2}}{p}\left\{\begin{array}{l} \omega_{1}\frac{\partial u_{k}}{\partial x_{k}}\bar{\bar{\frac{\partial u_{i}}{\partial x_{j}}}}+\omega_{2}\left[-\bar{\bar{\frac{\partial }{\partial x_{i}}\left(\frac{1}{\rho }\frac{\partial p}{\partial x_{j}}\right)}}-\bar{\bar{\frac{\partial u_{k}}{\partial x_{i}}\frac{\partial u_{j}}{\partial x_{k}}}}-2\bar{\bar{\bar{\bar{\frac{\partial u_{i}}{\partial x_{k}}}}\frac{\partial u_{k}}{\partial x_{j}}}}\right]+\omega_{3}\bar{\bar{\frac{\partial^{2}T}{\partial x_{i}\partial x_{j}}}}\\ +\omega_{4}\frac{1}{\rho T}\bar{\bar{\frac{\partial p}{\partial x_{i}}\frac{\partial T}{\partial x_{j}}}}+\omega_{5}\frac{R}{T}\bar{\bar{\bar{\bar{\frac{\partial T}{\partial x_{i}}\frac{\partial T}{\partial x_{j}}}}}}+\omega_{6}\bar{\bar{\bar{\bar{\frac{\partial u_{i}}{\partial x_{k}}\frac{\partial u_{k}}{\partial x_{j}}}}}} \end{array}\right\}$$
(6)$$q_{i}^{\left(2\right)}=\frac{\mu^{2}}{\rho }\left\{\begin{array}{l} \theta_{1}\frac{1}{T}\frac{\partial u_{k}}{\partial x_{k}}\frac{\partial T}{\partial x_{i}}+\theta_{2}\frac{1}{T}\left[\frac{2}{3}\frac{\partial }{\partial x_{i}}\left(T\frac{\partial u_{k}}{\partial x_{k}}\right)+2\frac{\partial u_{k}}{\partial x_{i}}\frac{\partial T}{\partial x_{k}}\right]+\theta_{3}\frac{1}{\rho }\frac{\partial p}{\partial x_{k}}\bar{\bar{\frac{\partial u_{k}}{\partial x_{i}}}}\\ +\theta_{4}\frac{\partial }{\partial x_{k}}\left(\bar{\bar{\frac{\partial u_{k}}{\partial x_{i}}}}\right)+\theta_{5}\frac{1}{T}\frac{\partial T}{\partial x_{k}}\bar{\bar{\frac{\partial u_{k}}{\partial x_{i}}}} \end{array}\right\}$$
where *R* is the gas constant and *T* is the temperature, and *μ* is the viscosity coefficient.

The viscous stress tensor *σ*_11_ can be written as:(7)$$\begin{array}{l} \sigma_{11}=-\mu (\frac{4}{3}u_{x}-\frac{2}{3}v_{y})+\frac{\mu^{2}}{p}(\alpha_{1}u_{x}^{2}+\alpha_{2}u_{y}^{2}+\alpha_{3}v_{x}^{2}+\alpha_{4}v_{y}^{2}+\alpha_{5}u_{x}v_{y}+\alpha_{6}u_{y}v_{x}+\alpha_{7}RT_{yy}\\ +\alpha_{8}RT_{yy}+\alpha_{9}\frac{RT}{\rho }\rho_{xx}+\alpha_{10}\frac{RT}{\rho }\rho_{yy}+\alpha_{11}\frac{RT}{\rho^{2}}\rho_{x}^{2}+\alpha_{12}\frac{RT}{\rho^{2}}\rho_{y}^{2}\\ +\alpha_{13}\frac{R}{T}T_{x}^{2}+\alpha_{14}\frac{R}{T}T_{y}^{2}+\alpha_{15}\frac{R}{\rho }T_{x}\rho_{x}+\alpha_{16}\frac{R}{\rho }T_{y}\rho_{y})\\ +\frac{\mu^{3}}{p^{2}}RT(\alpha_{17}u_{xxx}+\alpha_{17}u_{xyy}+\alpha_{18}v_{xxy}+\alpha_{18}v_{yyy}) \end{array}$$

The other viscous stress tensor *σ_ij_* is not introduced in detail here. For more specific information about the equations, please refer to the previous paper [13]. The coefficients of Burnett equations are shown in Table 1.

The compressible ideal gas model is:(8)p=ρRT.

To account for the nonequilibrium effects, the slip boundary conditions have been used. The classical slip boundary conditions are the Maxwell first-order slip boundary conditions:(9)$$u_{s}-u_{w}=\frac{2-\sigma_{v}}{\sigma_{v}}\lambda {\frac{du}{dy}|}_{w}+\frac{3}{4}\frac{\mu }{\rho T}{\frac{\partial T}{\partial x}|}_{w},$$
(10)$$T_{s}-T_{w}=\frac{2-\sigma_{T}}{\sigma_{T}}\frac{2\gamma }{Pr\left(\gamma +1\right)}\lambda {\frac{\partial T}{\partial y}|}_{w}$$
where *σ_v_* is the tangential moment accommodation coefficient and *σ_T_* is the thermal accommodation coefficient. *u_s_* and *T_s_* are the gas velocity and temperature, respectively; *u_w_* is the velocity of the wall, *Pr* is the Prandtl number and *γ* is the specific heat ratio.

In recent years, Sharipov et al. proposed the more correct form of boundary conditions derived from the kinetic theory [5,9]:(11)$$u_{s}=\sigma_{p}\frac{\mu }{p}v_{m}\left(\frac{\partial u}{\partial y}\right)|{}_{w}+\sigma_{T}\frac{\mu }{\rho T}\left(\frac{\partial T}{\partial x}\right)|{}_{w},$$
where *v_m_* = $\sqrt{2RT}$ is the most probable molecular speed, *T_w_* is the wall temperature, and *σ_p_* and *σ_T_* are the velocity slip and thermal slip coefficients.

Moreover, several experimental studies and theoretical researches have confirmed the range of the velocity and the thermal slip coefficient, which are helpful in dealing with rarefied gas flows [28,29,30].

For *Kn* > 0.1, experimental studies have shown that models based on the first-order boundary condition show considerable discrepancies against observed data [31]. In the present study, the second-order slip boundary conditions have been used. The classical slip boundary equation proposed by Beskok et al. [32] has been commonly adopted and written as:(12)$$u_{s}=\frac{1}{2}\left[u_{\lambda }+\left(1-\sigma_{v}\right)u_{\lambda }+\sigma_{v}u_{w}\right]+\frac{3\sigma_{v}}{8}\frac{\mu R}{P}{\frac{\partial T}{\partial x}|}_{w}$$
(13)$$T_{s}=\frac{\frac{2-\sigma_{T}}{Pr}\frac{\gamma }{\gamma +1}T_{\lambda }+\sigma_{T}T_{w}}{\sigma_{T}+\frac{\left(2-\sigma_{T}\right)}{Pr}}$$
where *u_λ_* and *T_λ_* are the velocity and temperature at a mean free path away from the wall, respectively. This kind of slip boundary condition corresponds to a high-order slip boundary condition by simply expanding *u_λ_* in terms of *u_s_*, using Taylor series expansion.

The forces acting on the particles are drag, Brownian, and thermophoretic forces. The governing equation using the Lagrangian tracking method can be written as [33]:(14)$$\frac{du_{p}}{dt}=F_{D}+\frac{g_{x}\left(\rho_{p}-\rho \right)}{\rho_{p}}+F_{B}+F_{T}$$
where *F_D_*, *F_B,_* and *F_T_* stand for the drag force, the Brownian force, and the thermophoretic force, respectively (subscript *p* represents particles).
(15)$$F_{D}=\frac{18\mu }{d_{p}^{2}\rho_{p}C_{c}}(u-u_{p})$$
where, *ρ* and *ρ_p_* are air density and particle density, respectively. *u* and *u_p_* are the air velocity and particle velocity, respectively. *C_c_* is the Cunningham slip correction factor, which can be calculated as [34]:(16)$$C_{c}=1+\frac{2\lambda }{d_{p}}\left(1.257+0.4e^{-\left(1.1d_{p}/2\lambda \right)}\right)$$

The random Brownian force per unit mass was given as a Gaussian white noise process following Kim and Zydney [35], using the expression:(17)$$F_{B}=\xi_{0}\sqrt{\frac{12\pi \mu d_{p}KT}{\Delta t}}$$
where *K* = (1.381 × 10^−23^ J K^−1^) is the Boltzmann constant, Δ*t* is the time elapsed, and *ξ_0_* is a zero-mean unit-variance-independent Gaussian random number.

The thermophoretic force is formulated as [36]:(18)$$F_{T}=4.5\pi \frac{\mu^{2}}{\rho }d_{p}\frac{1}{1+3Kn_{p}}\frac{\frac{C}{C_{p}}+2.48Kn_{p}}{1+2\frac{C}{C_{p}}+4.48\frac{C}{C_{p}}}\frac{\nabla T}{T}$$

The Knudsen number of the particle *K*n_p_ is defined as:(19)$$Kn_{p}=2\lambda /d_{p}$$
where *C* and *C_p_* are the thermal conductivity of air and the particle, respectively, and *μ* is the fluid viscosity.

## 3. Simulation Parameters and Solution Verification

In this study, the commercial software Fluent (version 16.0, Ansys, Inc., Canonsburg, PA, USA) with user-defined function has been acquired to compute the Burnett equations and particle dynamics equation. The governing partial differential equations were converted into algebraic equations by the finite volume method (FVM). The second-order central difference scheme was used for the diffusion terms while the convective terms were formulated by the Quick scheme of Spalding. The semi-implicit method for pressure-linked equations (SIMPLE) has been used for a two-dimensional model of the flow field in a microchannel [37]. The convergence criteria reduced the maximum relative error in the values of all dependent variables between two successive iterations below 10^–6^. The gas phase was a pressure-driven ideal nitrogen flow field. Figure 1 shows the schematic of the geometry and the parameters of the channel. *L* = 12 μm and *H* = 0.6 μm were the length and width of the channel, respectively. The flow velocity changed with the ratio *η* of the inlet pressure *p_1_* and the outlet pressure *p_2_*. The initial outlet pressure was 100 KPa and the temperature of gas or wall was 300 K. In this condition, the gas mean free path *λ* = 68.4 nm, so the flow Knudsen number was *Kn* = *λ*/*H* = 0.114.

In the past decades, the computational fluid dynamics-discrete particle method (CFD-DPM) has been successfully employed to investigate the particle-fluid flow in many engineering applications [38], so DPM was used to compute the movement of particles in this paper. Considering both precision and efficiency, the particle phase was taken into account in the following assumptions. Firstly, it was assumed that particles were smooth and spherical. Secondly, the inter-particle effects were ignored. Thirdly, a one-way coupling assumption was made, indicating that the fluid carried the particle, but the particle influence on the fluid was neglected. Fourthly, when the distance between the particle and the channel wall was less than its radius, we assumed that the particle was deposited on the wall surface. The distribution of particles was assumed uniform across the inlet. To ignore the interparticle collision, the particle volume concentration was less than 10%, which was reasonable considering the particle concentration and size in Sommerfeld’s study [39]. The nanoparticles were tracked until they deposited onto a wall or exited the channel.

In a microreactor or microfilter, the particle deposition is an important criterion of a microsystem process. So the nanoparticle deposition efficiency (*ξ*) on the surface was defined as [40]:(20)$$\xi =N_{deposition}/N_{total}\times 100\%$$
where *N_total_* was the number of the total particles released at the inlet, and *N_deposition_* was the deposition particle number on the wall. To improve the statistics of the particle moving, the result was averaged over more than 50 runs for every injection. The corresponding standard deviation of all of the depositional particle numbers was less than 5% in the same flow conditions.

Table 2 shows the results of a grid-independency study carried out at *η* = 1.5. From the result, it could be concluded that the 500 × 50 grid is enough for the following simulations, assuring iterative stability and solution precision.

Figure 2 compares the velocity distributions at a central axis of the microchannel with *η* = 1.5 solved by the Burnett equations against DSMC and N-S equations. Both of the boundary conditions of the Burnett equations and N-S equations are the second-order slip boundary conditions. The results of the Burnett equations were in good agreement with that of DSMC, which means the Burnett equations calculated the fluid field in the transition regime well. Comparing the results of N-S equations to the first two methods, the deviations increased with the gas flowing towards the outlet of the microchannel. So, the Burnett equations were more suitable than the N-S equations to study the fluid flow in the transition flow regime.

## 4. Results and Discussion

### 4.1. The Flow Field Characteristics and Influence on Particles Deposition

Figure 3 shows the contour in the microchannel at *η* = 1.5. The fluid flow was non-uniform at different lengths. The velocity of the centerline was higher than that near the wall and full-field gas speed increased along the exit direction. The main important properties of the gas flow in a microchannel are wall-adjacent slip velocity and the rarefaction effects. Figure 4 compares the velocity profiles of the slip condition to the no-slip flow conditions at *x*/*L* = 0.5. The two velocity distributions are parabolic shapes, and the main differences were the value of wall-adjacent velocity and the maximum speed in the central axis. The value of wall-adjacent velocity was nearly 45% of that of the inlet velocity. Figure 5 shows the variation in wall-adjacent slip velocity with the length and the pressure ratio *η* of the channel. The wall-adjacent slip velocity increased along the flow direction. The higher the mainstream velocity induced by larger *η*, the bigger the slip velocity was.

The rarefaction effect on the gas in the microchannel can be assessed by the density distributions. The results are observed as shown in Figure 6 by varying the pressure ratio of the inlet to outlet. The curvature of the density variation was almost linear when *η* = 1.5 and then increased with an increase in pressure ratio. Figure 6 also shows the change in the curvature of the density profile. According to Dongari et al. [42], when analyzing the gaseous slip flow in long microchannels, only the second-order slip model can capture this result while the classical first-order slip model cannot.

The flow field near the channel surface seriously affects the particle deposition process. Figure 7 shows the particles’ deposition efficiency *ξ* of *d_p_* = 10 nm under different boundary conditions with *η* = 1.5. The results show that the deposition efficiency of particles under the no-slip condition was about 2% higher than under the second-order slip conditions. Since the flow field wall-adjacent velocity of slip conditions in the *x*-direction was higher than that under the no-slip conditions, the diffusion effect was weaker, which impeded the particle deposition on the surface. Furthermore, the slip velocity reduced particle residence time in the channel which decreased the chance of particles touching the channel wall.

### 4.2. The Pressure Ratio Effect on the Particle Deposition Efficiency

Figure 8 shows that the deposition efficiency of particles of different sizes varied with the pressure ratio in a channel. Unlike with micro-size particles, applying Brownian force instead of gravity or inertial force resulted in nanoparticle deposition on the surface under constant temperature conditions. When the particle diameter was in the range of 10–50 nm, the particle deposition efficiency increased with the decreasing particle diameter and flow pressure ratio. The main factors were twofold: first, the larger particle meant weaker Brownian diffusion; second, a higher-flow pressure ratio induced shorter particle residence time. Comparing the results to those for a larger channel, the decreasing of gas density in the transition regime of a microchannel caused the drag force on the particles to recede [4], so the particles more easily touched the wall surface, which increased the particle deposition efficiency. As a result, particle deposition efficiency was higher in a rarefied gas channel than in a continuous gas channel with similar slip boundary conditions. Therefore, the efficiency of deposition increased with the higher Knudsen number of more rarefied gas.

To depict the particle deposition position and distribution, we divided the channel evenly by length into five parts and called them A, B, C, D, E from inlet to outlet. *ξ_R_* is the number of particles deposited at section A, B, C, D, or E, divided by the total quality of the deposition in the channel. Figure 9a–c show the particle deposition at different positions of the microchannel with the changing of particle diameter and pressure ratio. It can be seen that more than 50% of the deposited particles were in the first section (part A) while only a few deposited particles were in the last section (part E). *ξ_R_* of the particle with 10 nm, 30 nm, and 50 nm at *η* = 1.5 was 50.88%, 54.42%, and 59.32% in section A, respectively. *ξ_R_* of *d_p_* = 10 nm decreased obviously from 50.88% in section A to 5.75% in section E. The reason can be seen in Figure 2 and Figure 5. The gas velocity and the slip velocity increased along with the flow in the channel, so the drag force effect on the particle increased and accelerated the particle velocity. The larger particle velocity in *x*-direction resulted in easier escaping from the pipe and lower particle deposition efficiency on the channel surface.

Furthermore, it was also found that larger particles with higher inertia were more easily deposited in the forepart of the microchannel than smaller ones. The numbers of *ξ_R_* in section A increased from 59.32% to 75.81% when *d_p_* = 50 nm with *η* changing from 1.5 to 3, but increased from 50.88% to 58.14% when *d_p_* = 10 nm. Instead, the *ξ_R_* descended gradually in section E. When *η* = 2.5, the 50-nm-diameter particles would not deposit on the E section, and when *η* = 3.0, none of the 30-nm-diameter particles deposited on the last section. The result shows the significance of controlling the location of particle deposition. Above all, the competition among the random Brownian force, the drag force, and the particle inertia made the particle deposition vary in different locations.

### 4.3. The Temperature Effect on the Deposition Process

Particles in a flow field with a temperature gradient will experience thermophoretic force, and this force will lead the particles to move towards the direction of the lower temperature side. In order to solve the congestion problem in a micropipe, it is important to reduce the particle deposition, so an aerosol cooler than the channel surface was transported into the pipe. The inlet aerosol temperature was kept constant at 300 K and the temperature gradient was formed by changing the wall temperature. Therefore, the nanoparticles migrated away from the wall surface under the action of the thermophoretic force. Figure 10 shows the deposition efficiency of particles of different sizes under various temperature gradients at *η* = 1.5. The results show that with the increasing of the temperature gradient between aerosol and the wall, the decreasing deposition efficiency is obvious; e.g., the deposition efficiency of 30 nm particle decreased from 14.7% to 9.6%. The influence of the temperature gradient on larger particles was greater than the smaller ones. The deposition efficiency reduction of 50-nm-diameter particles was from 11.8% to 3.8%, significantly greater than that of 10-nm-diameter particles, which was reduced from 22.6% to 20.2%. This is the result of the competition between Brownian force, thermophoretic force, and particle inertia, which play major roles in the deposition process. Under the same temperature gradient, the 50-nm-diameter particle had a bigger thermophoretic force and a smaller Brownian diffusion than the 10-nm-diameter particle, which resulted in the 50-nm-diameter particle having less deposition opportunity.

The above research shows that the distribution of particle deposition on the surface is nonuniform and mainly concentrated in front of the channel. Hence, the same temperature gradient at different positions should have a different effect on particle deposition. So we changed the wall temperature of each section and obtained the particle deposition efficiencies respectively. Figure 11a–c shows the particle deposition quantity distribution when a temperature gradient existed in different areas. The subscript of Δ*T* is the position with a temperature gradient. The results show that the temperature gradient existed in section A of the pipe, where the most obvious reduction of the deposition efficiency happened. The section where the temperature gradient was closer to the end of the channel showed fewer impacts on particle deposition. When Brownian force, thermophoretic force, and drag force all affected the nanoparticles, the great change of the deposition efficiency happened when *d_p_* = 50 nm in this study, which means the influence on the larger particles was more obvious than that on the smaller ones.

## 5. Conclusions

In this study, numerical simulations acquiring augmented Burnett equations and a particle dynamics equation have been carried out to research the movement and deposition process of nanoparticles in a microchannel flow field. The effects of the slip boundary condition, inlet and outlet pressure ratio, particle diameter, and flow field temperature gradient on particle deposition efficiency have been analyzed. The competitive relationship among the Brownian force, the thermophoretic force, the drag force, and the particle inertia influence on the deposition efficiency of nanoparticles has been discussed.

The result shows the nanoparticle deposition on the channel surface is influenced by the combined effects of both flow field characteristics and forces acting on the nanosized particles. First, for the flow field, when the growth pressure ratio of inlet and outlet increases the slip velocity, nanoparticle deposition efficiency decreases. However, the efficiency of deposition increases with a higher Knudsen number of more rarefied gas. Second, for the effect of the forces on the particle, the Brownian force leads the smaller size particle to move in a more vigorous but random direction, which strengthened these particles’ deposition on the surface. However, the thermophoretic force in the direction of the lower temperature-side had more obvious effects on the larger nanoparticles than the smaller ones. Finally, adjusting the aerosol flow rate and temperature gradient can change the particle deposition efficiency and location in a microchannel on various levels for different particle sizes.

## Figures and Tables

**Figure 1 micromachines-12-00047-f001:**
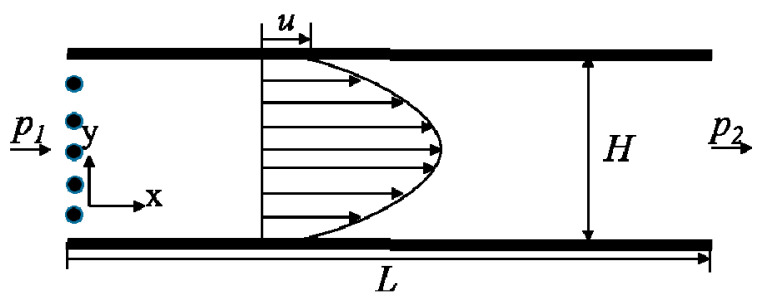
Schematic of the geometry of the microchannel.

**Figure 2 micromachines-12-00047-f002:**
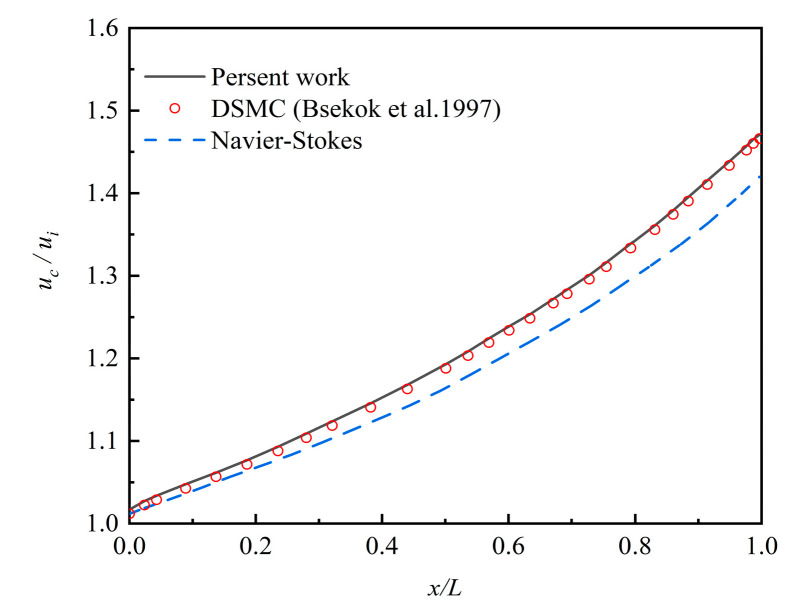
The dimensionless velocity distribution along the centerline of the microchannel at *η* = 1.5.

**Figure 3 micromachines-12-00047-f003:**
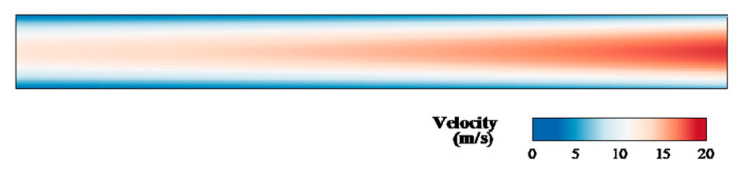
The velocity contour in the microchannel at *η* = 1.5.

**Figure 4 micromachines-12-00047-f004:**
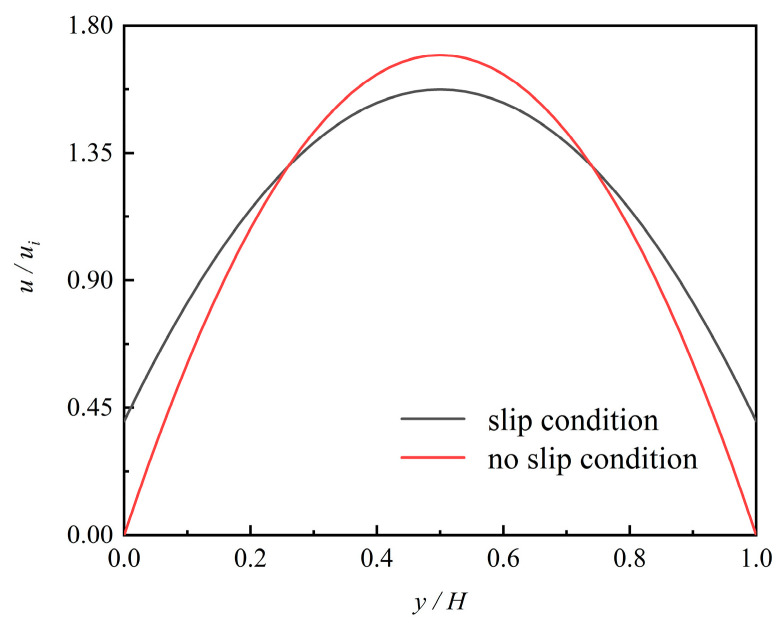
Velocity profiles for the slip and no-slip flow conditions at *x*/*L* = 0.5.

**Figure 5 micromachines-12-00047-f005:**
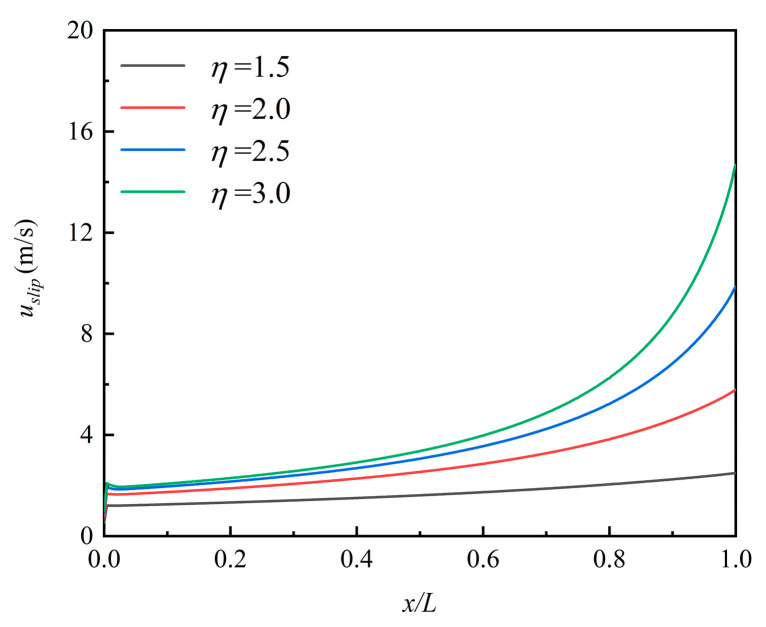
The variation of the wall-adjacent slip velocity with *x* at different *η*.

**Figure 6 micromachines-12-00047-f006:**
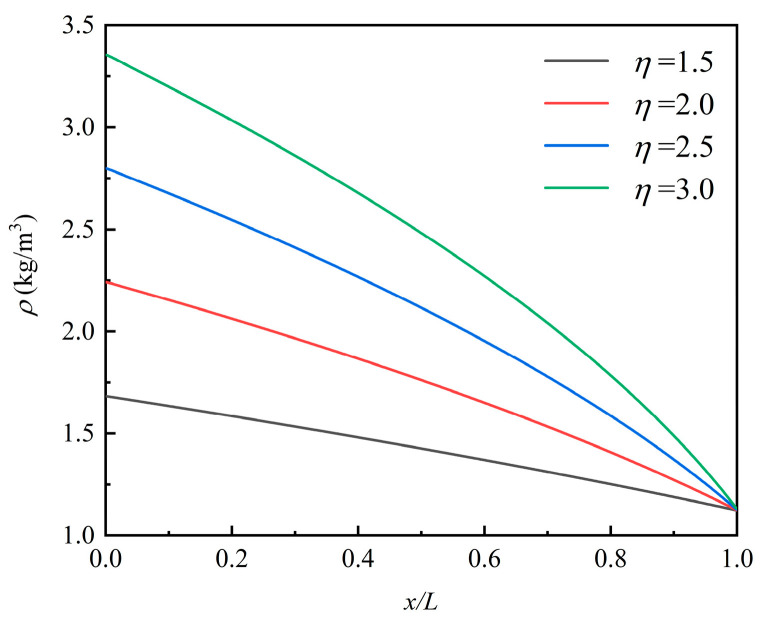
The density distributions of the airflow in the channel under different *η*.

**Figure 7 micromachines-12-00047-f007:**
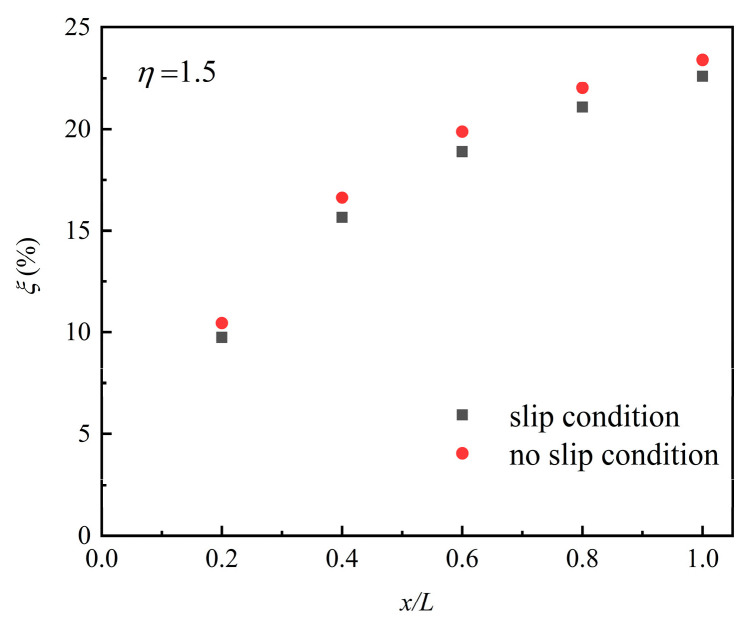
The deposition efficiency of particles with *d_p_* = 10 nm under different boundary conditions at *η* = 1.5.

**Figure 8 micromachines-12-00047-f008:**
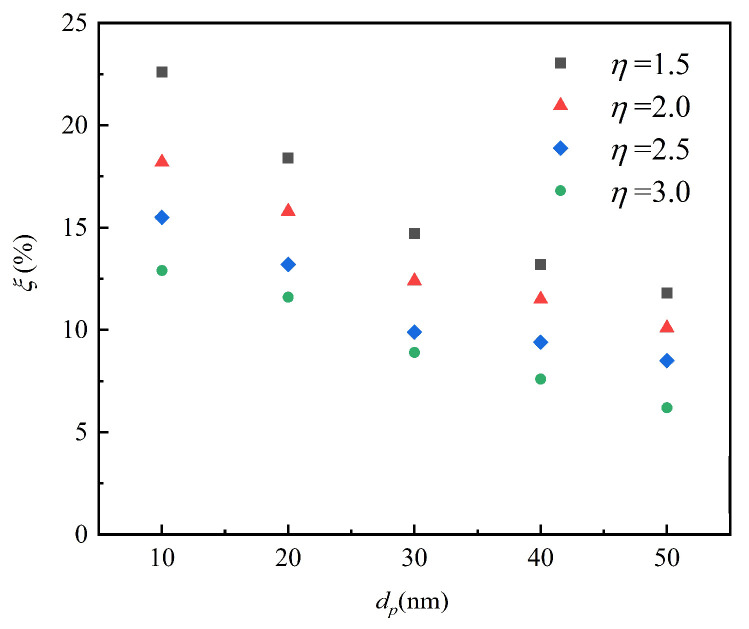
The particle deposition rate changing with the diameter at different pressure ratios.

**Figure 9 micromachines-12-00047-f009:**
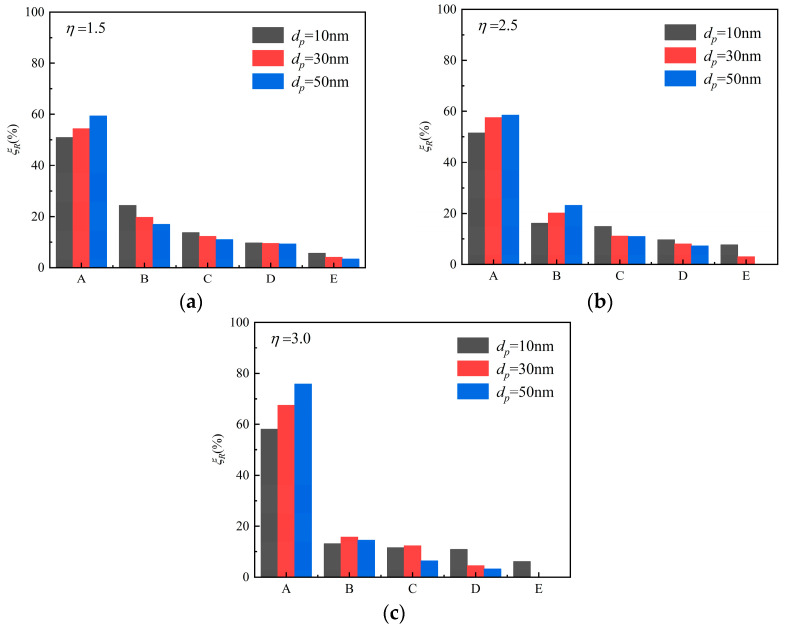
The particle deposition distribution along the length of the channel with different particle diameters and airflow pressure ratios. (**a**) *η* = 1.5; (**b**) *η* = 2.5; (**c**) *η =* 3.0.

**Figure 10 micromachines-12-00047-f010:**
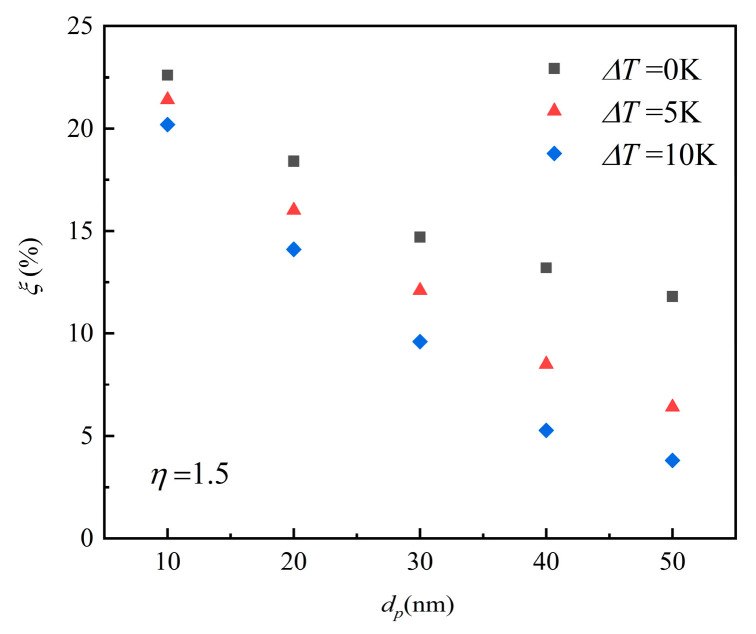
The particle deposition efficiency with different temperature gradients at *η* = 1.5.

**Figure 11 micromachines-12-00047-f011:**
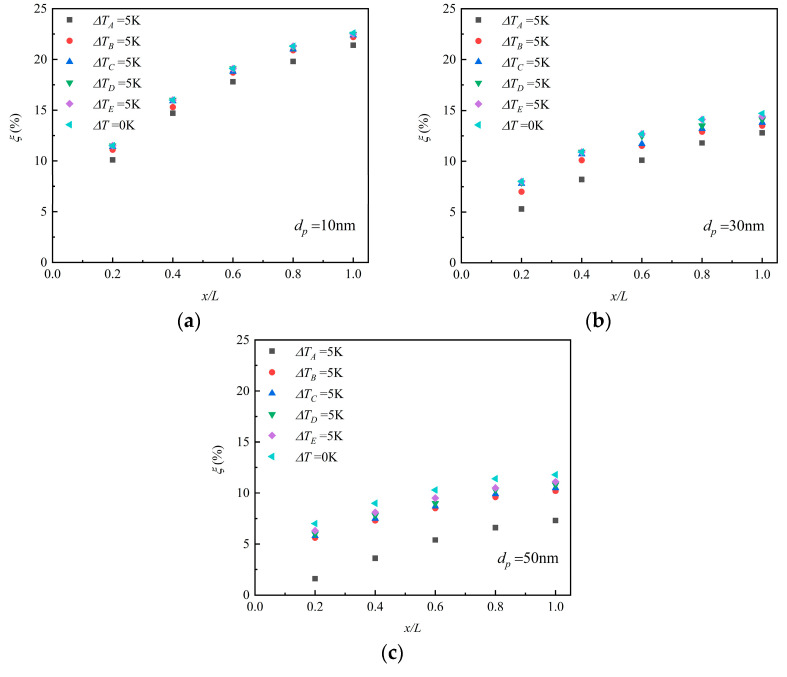
The particle deposition efficiency with different temperature gradients at *η* = 1.5. (**a**) *d*_p_ = 10 nm; (**b**) *d*_p_ = 30 nm; (**c**) *d*_p_ = 50 nm.

**Table 1 micromachines-12-00047-t001:** Values of the Burnett coefficients.

Coefficients	Maxwell Molecules	Hard-Sphere Molecules
*ω* _1_	10/3	4.056
*ω* _2_	2	2.028
*ω* _3_	3	2.418
*ω* _4_	0	0.681
*ω* _5_	3	0.219
*ω* _6_	8	7.424
*θ* _1_	75/8	11.644
*θ* _2_	−45/8	−5.822
*θ* _3_	−3	−0.393
*θ* _4_	3	2.418
*θ* _5_	117/4	25.157

**Table 2 micromachines-12-00047-t002:** Gas velocity at *x* = *L*/2, *y* = *H*/2 for the grid-independency verification carried out at *η* = 1.5.

Case	Total Cell Number	Velocity	Relative Error from DSMC [41] (%)
Case 1	200 × 20	14.6839	1.47
Case 2	300 × 30	14.8025	0.67
Case 3	400 × 40	14.8582	0.30
Case 4	500 × 50	14.8913	0.08
Case 5	600 × 60	14.8956	0.05

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
