# Peer review of "Numerical Study of Nanoparticle Deposition in a Gaseous Microchannel under the Influence of Various Forces"

_micromachines, 2021, doi:10.3390/mi12010047_

Round 1
Reviewer 1 Report
my comments are in attached file

Reviewer 2 Report
This work considers the deposition of nanoparticles in a microchannel. The major significance is that the coupling between rarefied gas dynamics and the movement of the particles. Augmented Burnett equations are solved to provide the gas for the particles. The simulation is performed at a variety of parameters settings.
I'm generally satisfied this work. The paper is well structured, and there are sufficient results to support the authors' statements. The major problem I found is that some details of the model and the computation are missing:
- As there are many different versions of Burnett equations, the authors should explain why the augmented Burnett equations are chosen. The origin of the augmented Burnett equations [X. Zhong, R.W. MacCormack, and D.R. Chapman, Stabilization of the Burnett Equations and Applications to Hypersonic Flows. AIAA Journal 31, 1036 (1993)] should also be cited.
- The coefficients in the Burnett equations depend on the type of gas molecules. This is not given in the paper.
- How is the deposition of the particles detected numerically? Are the particles removed from the simulation after deposition?
- The authors should provide some information on the numerical method. What is the formal numerical order of the scheme? Does the numerical result agree with the numerical order?
- The readers may be interested in seeing the complete flow structure in the channel. I suggest the authors provide a 2D plot to show the velocity field for some parameter settings.
- It is unclear to me how the simulation is done. What is the initial condition of the flow? Or the authors first find the steady state of the flow and then add particles? If a dynamical flow is simulated, what is the final time of the simulation?
Round 2
Reviewer 1 Report
Referee comments
The authors of the manuscript ”Numerical study of nanoparticles depo- sition in a gaseous microchannel under the influence of various forces” have improved the manuscript.
The referee has some the following points to be corrected:
1. On page 1 the Ref. [5] does not mentioned correctly.
2. On page 3, the bar and double bars in expressions of the shear stress and heat flux must be explained, Eqs. (5) and (6).
To conclude, It is proposed the publication of this manuscript after the corrections mentioned above.